# Peer review of "Robotic Hysterectomy with Ureter Identification and Uterine Artery Ligation for Benign Gynecological Conditions: An Early-Year Single-Center Experience"

_diagnostics, 2023, doi:10.3390/diagnostics13101809_

Round 1

Reviewer 1 Report

Robotic surgery has gained a special place in minimally invasive surgery for benign disease in the gynecologic field. Many studies have already assessed the costs, the advantages, and the length of the hospital stay compared to laparoscopic procedures.

The current study discloses in the title the merits of using ureter identification and preventive uterine artery ligation technique. The article, as I see it, has many flaws.

1. The Introduction section is way too long. It is a medium-sized Discussions section, and it debates mainly the advantages of the robotic approach for hysterectomies.

2. It is not quite clear the purpose of the study - to demonstrate the need and benefits of ureteral dissection or to underline the advantages of the robotic approach.

3. Larger studies are available, and newer ones already debate the advantages of the robotic approach for hysterectomies.

4. The references are quite old, as I said in point 3, newer ones are available (such as doi: 10.1136/ijgc-2020-001611. Epub 2020 Aug 30,  doi: 10.1007/s11701-020-01074-7. Epub 2020 Apr 6, 2022 Dec 23;20(1):234. doi: 10.3390/ijerph20010234.)

5. The Conclusions section includes almost everything. Perhaps the Limitations of this study should be described separately, and once the authors figure out which is the main focus of this article, the Conclusions should refer only to this main research goal.

The English Language is acceptable.

Author Response

We would like to express our gratitude for your feedback on the manuscript. We have carefully reviewed and revised the document based on the insightful comments provided by the reviewers. We truly appreciate their valuable and constructive feedback. We sincerely hope that the revisions we have made meet your expectations. If there are any additional suggestions or feedback, please do not hesitate to let us know. We are committed to making any necessary modifications to ensure the quality of the manuscript.

Reviewer 2 Report

Overall, this is an interesting paper, which describes a single-center clinical experience on robotic hysterectomy with standardized clinical steps of ureter identification and uterine artery ligation. Since he approval of the Da Vinci robotic surgery system has been used extensively in certain gynecologic centers to treat benign uterine conditions. In the past years laparoscopic total hysterectomy (TLH) took over the place of open laparotomy and has become the gold standard of hysterectomy. However, avoiding ureteral injury and reducing the blood loss remained major challenges, thus, different techniques has been introduced to minimize the risks. The described standardized technique used by the authors is one of the most frequently applied method at TLH today, which can be further improved by the robotic technique. Most of results of the authors are impressive, but the study has several limitations as well. 

Advantages of the study is the standardization of the operative technique and the single center setting, which results in a uniform clinical practice and reliable outcome data of surgical technical and clinical details.

Among the limitations are the small patient number and the difficult adaptation of an almost ten years old dataset (2014-2015) for the current clinical practice. We must note, that the robotic hysterectomy has been intensively improved in the past years, as well as the number of the operations. Thus, I would recommend indicating this fact in the title of the paper, eg. ‘early experiences of a single medical center’. Further it is not clear, how the patients were recruited in this study, it is hardly likely, that only 49 hysterectomies were performed in that 20 months of the study period. Regarding the average and SD of the uterine weight and the BMI, the uterine size were rather uniform and body constitutions were ideal. What were the preoperative inclusion and exclusion criteria for robotic hysterectomy, or were the patients consecutively enrolled? Since it it is indicated in the Discussion (line 273), that majority of the patients underwent robotic subtotal hysterectomy (i.e. a cervix was not removed, which clearly affects the complication risk and blood loss rate), the reasons and the percentage of those has to be clearly shown, as well as the removal technique of the uterine corpus (morcellization or mini-laparotomy?). The total and subtotal hysterectomies should be analyzed separately. In the reflection of the above mentioned and the preop Hgb I find the blood transfusion need (14,3%) too much, it needs to be explained. My major problem that since there is no control group in the study (TLH), the results are simply descriptive and hardly can be translated to clinical practice. 

In the Discussion (line 218-229) the technical details of the ureter dissection and uterine artery ligation is described, this section seems better fit in the Methods part. Here is concluded, that this approach “resulted in a significant reduction in blood loss and transfusion rates…” – but comparing to what?

My overall impression that this paper needs extensive reconsiderations, including the setting of a control group, clarifying the inclusion and exclusion criteria, and treating the total and subtotal hysterectomies separately. 

Author Response

We have taken great care to revise the manuscript in response to the comments provided by the reviewers. We sincerely appreciate their valuable and constructive feedback. Our team hopes that the revisions we have made meet their expectations. If there are any additional suggestions or feedback, please do not hesitate to let us know. We are committed to making any necessary modifications to ensure the quality of the manuscript.

Round 2

Reviewer 1 Report

I read with great interest the revised form of the article. Some of the previous requests were amended, but a few were not. 

Point 1 As I see it, the Conclusions section is too long. It should contain a shorter message, not discussions material.

Point 2 In the Discussions Section, lines 303-308, The authors declare that 'the majority of their study population underwent subtotal hysterectomy'. This statement is important for the overall message. Please state these numbers within the results section. In light of this particular surgical technique, which might be the point of dissecting the ureter if the subtotal hysterectomies traditionally do not involve ligating the uterine artery right above the ureter?

The overall quality of the English language is satisfactory.

Author Response

Dear Reviewer,

We sincerely appreciate your time and effort in reviewing our manuscript and providing us with your feedback. We are grateful for your positive comments on our innovative technical approach to improving the clinical outcome and experience of robotic surgery.

Your valuable feedback has been instrumental in strengthening our study, and we are grateful for your constructive criticism. We have carefully considered all of your comments and have made significant revisions to our manuscript to address your concerns.

We are humbled by your encouragement and support, and we hope that our research contributes to the growing body of literature on this topic. We look forward to hearing any further suggestions or feedback that you may have.

Once again, thank you for your time and expertise in reviewing our manuscript. Your contribution has been invaluable to the improvement of our work.

Reviewer 2 Report

Thank you for the detailed answers to my concerns, as well as the corrections and the add-in sentences in the revised paper. I think, that these improved the originality and quality of this research, which otherwise reflects innovative technical approach to improve the clinical outcome and experience, from the early years of robotic surgery. 

Author Response

Dear Reviewer :

We sincerely appreciate your time and effort in reviewing our manuscript and providing us with your feedback. We are delighted to hear that you found our revisions to have improved the originality and quality of our research, and we are grateful for your positive comments on our innovative technical approach to improving the clinical outcome and experience of robotic surgery.

Your valuable feedback has been instrumental in strengthening our study, and we are grateful for your constructive criticism. We have carefully considered all of your comments and have made significant revisions to our manuscript to address your concerns.

We are humbled by your encouragement and support, and we hope that our research contributes to the growing body of literature on this topic. We look forward to hearing any further suggestions or feedback that you may have.

Once again, thank you for your time and expertise in reviewing our manuscript. Your contribution has been invaluable to the improvement of our work.